# NKG2A Expression among CD8 Cells Is Associated with COVID-19 Progression in Hypertensive Patients: Insights from the BRACE CORONA Randomized Trial

**DOI:** 10.3390/jcm11133713

**Published:** 2022-06-27

**Authors:** Renata Moll-Bernardes, Sérgio C. Fortier, Andréa S. Sousa, Renato D. Lopes, Narendra Vera, Luciana Conde, André Feldman, Guilherme Arruda, Mauro Cabral-Castro, Denílson C. Albuquerque, Thiago C. Paula, Thyago Furquim, Vitor A. Loures, Karla Giusti, Nathália Oliveira, Ariane Macedo, Pedro Barros e Silva, Fábio De Luca, Marisol Kotsugai, Rafael Domiciano, Flávia A. Silva, Mayara F. Santos, Olga F. Souza, Fernando A. Bozza, Ronir R. Luiz, Emiliano Medei

**Affiliations:** 1D’Or Institute for Research and Education, Rio de Janeiro 22281-100, Brazil; renata.moll@idor.org (R.M.-B.); sergiofortier@gmail.com (S.C.F.); andreasilvestre0203@gmail.com (A.S.S.); renato.lopes@duke.edu (R.D.L.); andre.feldman@hotmail.com (A.F.); guilherme.arruda@saoluiz.com.br (G.A.); denilson.albuquerque@idor.org (D.C.A.); sopconstar@gmail.com (T.C.P.); thyago.furquim@saoluiz.com.br (T.F.); vitor.aloures@saoluiz.com.br (V.A.L.); karlagouvea@yahoo.com.br (K.G.); nathalia.m.oliveira@hospitalvillalobos.com.br (N.O.); arianevsm@yahoo.com.br (A.M.); deluca.cardio@gmail.com (F.D.L.); marisol.dorizo@saoluiz.com.br (M.K.); rafael.domiciano@saoluiz.com.br (R.D.); famoroso.ms@gmail.com (F.A.S.); mayara.fraga@idor.org (M.F.S.); olga.souza@rededor.com.br (O.F.S.); bozza.fernando@gmail.com (F.A.B.); ronir@iesc.ufrj.br (R.R.L.); 2Pathological Anatomy Laboratory, Rede D’Or São Luiz, São Paulo 04321-120, Brazil; 3Evandro Chagas National Institute of Infectious Diseases, Oswaldo Cruz Foundation, Rio de Janeiro 21040-360, Brazil; 4Duke Clinical Research Institute, Duke University Medical Center, Durham, NC 27710, USA; 5Brazilian Clinical Research Institute, São Paulo 01404-000, Brazil; drpedrobarros80@gmail.com; 6Institute of Biophysics Carlos Chagas Filho, Federal University of Rio de Janeiro, Rio de Janeiro 21941-170, Brazil; vera.narendra@gmail.com (N.V.); lucianaconde@biof.ufrj.br (L.C.); 7São Luiz Anália Franco Hospital, São Paulo 03313-001, Brazil; 8São Luiz São Caetano Hospital, São Caetano do Sul 09531-205, Brazil; 9Institute of Microbiology Paulo de Góes, Federal University of Rio de Janeiro, Rio de Janeiro 21941-902, Brazil; maurojorge@micro.ufrj.br; 10Cardiology Department, Rio de Janeiro State University, Rio de Janeiro 20551-030, Brazil; 11São Luiz Jabaquara Hospital, São Paulo 04321-120, Brazil; 12Sino Brasileiro Hospital, Osasco 06016-050, Brazil; 13Villa Lobos Hospital, São Paulo 03184-020, Brazil; 14Santa Casa of São Paulo, São Paulo 01221-010, Brazil; 15São Luiz Morumbi Hospital, São Paulo 05605-050, Brazil; 16Copa Star Hospital, Rio de Janeiro 22031-012, Brazil; 17Public Health Studies Institute—IESC, Federal University of Rio de Janeiro, Rio de Janeiro 21941-592, Brazil; 18National Center for Structural Biology and Bioimaging, Federal University of Rio de Janeiro, Rio de Janeiro 21941-902, Brazil

**Keywords:** COVID-19, NKG2A, HLA-DR, T cell, hypertension, immune response

## Abstract

Cardiovascular comorbidities and immune-response dysregulation are associated with COVID-19 severity. We aimed to explore the key immune cell profile and understand its association with disease progression in 156 patients with hypertension that were hospitalized due to COVID-19. The primary outcome was progression to severe disease. The probability of progression to severe disease was estimated using a logistic regression model that included clinical variables and immune cell subsets associated with the primary outcome. Obesity; diabetes; oxygen saturation; lung involvement on computed tomography (CT) examination; the C-reactive protein concentration; total lymphocyte count; proportions of CD4+ and CD8+ T cells; CD4/CD8 ratio; CD8+ HLA-DR MFI; and CD8+ NKG2A MFI on admission were all associated with progression to severe COVID-19. This study demonstrated that increased CD8+ NKG2A MFI at hospital admission, in combination with some clinical variables, is associated with a high risk of COVID-19 progression in hypertensive patients. These findings reinforce the hypothesis of the functional exhaustion of T cells with the increased expression of NKG2A in patients with severe COVID-19, elucidating how severe acute respiratory syndrome coronavirus 2 infection may break down the innate antiviral immune response at an early stage of the disease, with future potential therapeutic implications.

## 1. Introduction

Risk factors such as hypertension, diabetes, older age and obesity have been associated with worse prognosis in patients with coronavirus disease 2019 (COVID-19). Among these, hypertension is particularly important due to its high prevalence and global burden, however, the mechanisms of this association have not been fully clarified [1,2,3,4,5].

Previous studies emphasized the role of inflammation in the pathogenesis of hypertension [6]. Various subsets of immune cells such as B and T lymphocytes that are involved in innate and adaptive immune responses are implicated in vascular injury in hypertension [7]. Likewise, a dysregulated immune response has been described as a hallmark of severe COVID-19 and is associated with the progression to acute respiratory distress syndrome and death [8,9,10,11]. Thus, research groups have made significant efforts to determine the role of immune cells in COVID-19 [12,13,14].

One of the most recurrent events that is observed in COVID-19 is robust lymphopenia [15]. Decreases in all lymphocytes, including T, B, and natural killer (NK) cells, have been reported in patients with this disease [12]. This immunological profile is strongly associated with disease severity [16]. In addition, Zheng et al. [17] reported an increased expression of CD94/natural killer group 2 member A (NKG2A) receptor in patients that were infected with SARS-CoV2.

NKG2A is a member of the C-type lectin-like receptor superfamily, and this inhibitory receptor is known to be present in several immune cells, including CD8 cells. These lymphocytes mediate immunosurveillance against viral infection and virus-induced neoplasia [18,19]. According to Zheng [17] an increased expression of NKG2A may be associated with COVID-19 progression, and the downregulation of NKG2A expression may correlate with disease recovery. These findings suggest that the functional exhaustion of cytotoxic lymphocytes is associated with severe acute respiratory syndrome coronavirus 2 (SARS-CoV-2) infection. Hence, SARS-CoV-2 infection may break down antiviral immunity at an early stage.

To our knowledge, there is no previous publication analyzing immune cell subsets in such a large number of hypertensive patients with COVID-19. In this study, we aimed to better understand which immune cell subtypes are associated with more severe disease in this group of patients and explore the physiopathology of the immune response in COVID-19.

## 2. Materials and Methods

### 2.1. Population and Design

Patients that were included in this study were part of the BRACE CORONA trial [20], an investigator-initiated, phase-IV, multicenter, open-label, registry-based randomized trial involving 659 patients on angiotensin-converting enzyme inhibitors (ACEIs)/angiotensin II receptor blockers (ARBs) with confirmed COVID-19, who were hospitalized at 29 centers in Brazil. The present study was a secondary analysis conducted with blood samples from 156 hospitalized patients with hypertension that were enrolled consecutively in the trial at six centers in the state of São Paulo, Brazil. The samples were collected within 72 h of hospital admission, after COVID-19 diagnosis confirmation, between 21 May and 27 June 2020. The trial protocol [21] was approved by the Brazilian Ministry of Health’s National Commission for Research Ethics and by the institutional review boards or ethics committees of participating sites. All patients provided informed consent before enrollment.

Patients that were eligible for the BRACE CORONA trial were aged ≥18 years and were on treatment with ACEI/ARB at baseline. Patients with clinical indications for ACEI/ARB treatment termination on admission, such as hypotension, acute kidney injury, and/or shock, were excluded. Patients on mechanical ventilation and those with hemodynamic instability, acute renal failure, or shock on admission were also excluded [20] The inclusion and exclusion criteria are provided in detail in the Appendix A.

### 2.2. Outcomes

The primary outcome was progression to severe disease, according to the modified World Health Organization (WHO) Ordinal Scale for Clinical Improvement, during hospitalization. The scores on this scale range from 0 (no evidence of infection) to 8 (death); disease was classified as non-severe (mild to moderate, scores of 3–5), ranging from the lack of need for oxygen therapy to conditions requiring noninvasive ventilation, and severe (scores of 6–8), including disease requiring mechanical ventilation, inotropic support, and/or renal replacement therapy, and that caused death (Appendix A) [22]. Secondary outcomes were the lengths of stay (LOSs) in the hospital and intensive care unit (ICU), acute myocardial infarction, new or worsening heart failure, hypertensive crisis, transient ischemic attack, stroke, myocarditis, pericarditis, arrhythmia requiring treatment, and thromboembolic events.

### 2.3. Clinical and Laboratory Data

Baseline patient characteristics were assessed on admission and included sex, age, and oxygen saturation, as well as the extent of lung involvement on computed tomography (CT) examination and C-reactive protein (CRP) level, measured by latex-enhanced immunoturbidimetric assay. Comorbidities, including obesity, diabetes, asthma, chronic obstructive pulmonary disease, dyslipidemia, and coronary artery disease, were also recorded on admission. The criteria that were used for the identification of these complications have been provided in the BRACE-CORONA trial report [21].

### 2.4. Peripheral Blood Mononuclear Cell Isolation

Blood samples were collected from the patients into ethylenediamine-tetraacetic acid (EDTA) tubes (BD Vacutainer^®^ spray-coated K2EDTA Tube, Becton, Dickinson and Company, New Jersey, USA), which were centrifuged at 1500× *g* for 15 min at 21 °C. Lymphocytes and monocytes were quantified in an automized ABX Micros 60 system (Horiba Medical, Montpellier, France) using photometry. To obtain peripheral blood mononuclear cells (PBMCs), density gradient centrifugation (Ficoll-Paque, GE Healthcare, Piscataway, NJ, USA) was performed. The samples were then processed. One milliliter of phosphate-buffered saline (PBS) was added to each original tube. To a separate 50-mL Falcon tube, 9 mL of Ficoll-Paque PLUS (GE Healthcare) and the same amount of PBS were added. The blood solution was transferred to the Ficoll-Paque tube slowly to separate the layers. This was centrifuged continuously at 18–20 °C and 830× *g* for 15 min. Then, the buffy coat was transferred to a new 50-mL Falcon tube and PBS was added to a volume of 10 mL, followed by homogenization. The tube was centrifuged continuously at 830× *g* for 8 min, the supernatant was removed, and the pellet was resuspended by tapping the tube with PBS to a volume of 10 mL. The tube was then centrifuged at 890× *g* for 5 min, the supernatant was removed, and the pellet was resuspended by tapping the tube with Roswell Park Memorial Institute (RPMI) medium to a volume of 10 mL. The tube was centrifuged at 890× *g* for 5 min, the supernatant was removed, and the process was repeated. The obtained PBMCs were frozen at –80 °C in a solution containing 400 µL of fetal bovine serum (FBS), 400 µL of RPMI medium, and 200 µL of dimethyl sulfoxide.

The cryopreserved PBMC samples were removed from the freezer and thawed rapidly in a 37 °C water bath. They were then transferred to a 15-mL Falcon tube, and RPMI medium with 10% FBS was added drop by drop while mixing slowly. One to two drops were added every 10 s; when the solution reached the volume of 2 mL, RPMI medium was added to a final volume of 10 mL. The cells were washed twice with RPMI medium with 10% FBS with centrifugation at 840× *g* for 5 min at room temperature.

### 2.5. Flow Cytometry

To quantify the immune cell populations in the PBMCs, 1 × 10^6^ cells were stained with various combinations of fluorophore-conjugated antibodies (Appendix A). After incubation and washing, the samples were analyzed by flow cytometry in a BDFACS Canto II apparatus (Becton Dickinson, San Jose, CA, USA) using the BioConductor R packages (bioconductor.org; accessed on 15 October 2021) at the Pathology Laboratory of Rede D’Or São Luiz hospitals. The technical procedures and flow cytometry followed the laboratory’s previously validated standards. All fluorochrome-conjugated antibodies were validated before use, with determination of the reaction specificity and best volume for antigen–antibody-binding saturation (titration). The following profiles were quantified: total monocytes; total lymphocytes; B lymphocytes; T lymphocytes; NK cells; CD4+ T cells; CD8+ T cells; CD4/CD8 relationship; CD8+ CD38+ T cells (percentage and CD38 mean fluorescent intensity–(MFI)); CD8+ HLA-DR+ T cells (percentage and HLA-DR MFI); CD8+ NKG2A+ T cells (percentage and NKG2A MFI); CD8+ HLA-DR+ CD38- T cells; CD8+ HLA-DR+ CD38+ T cells; CD8+ HLA-DR- CD38+ T cells; and CD8+ HLA-DR- CD38- T cells. The approximate membrane expression (MFI) of the antigens HLA-DR and CD38, important markers of T-cell activation, and of NKG2A, an inhibitory receptor of T cells, were evaluated to improve the understanding of lymphocyte properties.

### 2.6. Statistical Analysis

Continuous variables were described as medians, means, and standard deviations; categorical variables were characterized as proportions. For the primary outcome, 95% confidence intervals (CIs) were calculated. Fisher’s exact test was used to detect the statistical associations between the outcome and categorical clinical variables. For continuous variables, receiver operating characteristic curves were used to evaluate associations with disease progression. Biomarkers that were associated significantly with the primary outcome were dichotomized using cutoff points of 90% sensitivity.

To better understand the associations of immune cell profiles and clinical variables with the risk of COVID-19 progression, a forward stepwise predictive multiple model was proposed. Due to the large number of biomarkers, a preliminary univariate analysis selected those that were associated with the outcome with areas under receiver operating characteristic curves (AUCs) > 0.65. These selected biomarkers with higher potential predictive value were combined with clinical variables and included in the model. The significance level for the entry and removal of the variables that were selected by automatic regression was set to 5%. Beta coefficients and odds ratios (ORs) were calculated for all variables in each step of the model to quantify associations with the outcome. The goodness of fit of the final model was evaluated by the Hosmer–Lemeshow test. Predicted probabilities of the primary outcome were estimated using variables with significant associations in the final model. All analyses were performed using SPSS software (version 24.0; IBM Corporation, Armonk, NY, USA).

## 3. Results

The mean patient age was 53.8 ± 12.2 years, and 54 (34.6%) patients were female. Among the 156 hypertensive patients, 12.2% were using ACEIs and 87.8% were using ARBs. Regarding comorbidities, 81 (51.9%) had obesity, 40 (25.7%) had diabetes, and 25 (16%) had dyslipidemia; two (1.3%) patients had heart failure and three (1.9%) had coronary artery disease. Asthma was reported in three (1.9%) cases, and chronic pulmonary and chronic renal disease were each reported in 1.3% of cases (Table 1). Five of 156 patients (3.2%) were using glucocorticoids before hospital admission. Data on all comorbidities are provided in Appendix A.

All patients had non-severe COVID-19 (WHO scores of 3–5) on admission. Cough (62.2%), fever (57.1%), myalgia (47.4%), shortness of breath (44.2%), fatigue (43.6%), and headache (32.1%) were the most common symptoms at presentation (Appendix A). The mean interval from symptom onset to hospital presentation was 5.6 ± 3.1 days, and 19.2% of patients had ≤93% oxygen saturation on admission. On chest CT examinations, 59.6% of patients showed ≤25% lung involvement, 35.9% showed 26–50% involvement, and 4.5% showed >50% lung involvement. Thirty-two (20.5%) cases had significant pulmonary involvement (oxygen saturation ≤93% and/or >50% lung involvement on CT) at admission (Appendix A).

### 3.1. Outcomes

Eleven (7.1%; 95% CI, 3.8–11.9%) patients progressed to severe disease during hospitalization, including three (1.9%) in-hospital deaths (Appendix A). Progression to severe disease was associated with obesity (*p* = 0.010), diabetes (*p* < 0.001), and oxygen saturation ≤ 93% or lung involvement >50% (*p* < 0.001) on admission, but not with age or sex (Table 1). Baseline patient characteristics and the most relevant comorbidities in patients that progressed or not to severe disease were presented in Table 1.

The mean hospital LOS was 9.1 ± 6.8 days. In total, 111 (71.2%) patients were admitted to the ICU; the mean ICU LOS was 7.6 ± 6.8 days (Appendix A). According to the report on the BRACE-CORONA trial [21], the mean numbers of days spent alive and out of hospital did not differ among patients that were hospitalized with mild to moderate COVID-19, according to ACEI/ARB discontinuation or continuation. The most common complications were acute renal injury (8.3%), transient ischemic attack (4.5%), sepsis (3.8%), hemodynamic decompensation (3.2%), and hypertensive crisis (3.2%; Figure 1).

### 3.2. Immune Cell Profiles and Biomarkers

Blood samples were collected at a mean of 2.7 days after hospitalization. The mean CRP concentration and proportion of CD8+ T cells were elevated, and the total lymphocyte count was reduced in patients who progressed to severe disease (Table 2). Counts for all T-cell subsets were lower in the severe disease group; the reduced CD4+ T-cell count led to a reduction in the CD4/CD8 ratio. Patients who progressed to severe disease showed increases in the MFIs of HLA-DR and NKG2A in CD8+ T cells (Table 3). However, CD38 marker expression was similar in the two groups (Table 2).

The CRP concentration, total lymphocyte count, percentage of CD4+ T cells, and CD4/CD8 ratio were associated with disease progression (AUCs, 0.827, 0.702, 0.718, and 0.692, respectively). The percentage of CD8+ T cells and MFIs of HLA-DR and NKG2A in CD8+ T cells were also associated with disease progression (AUCs, 0.659, 0.670, and 0.650, respectively).

### 3.3. Predictive Model

The variables that were associated with disease progression in a preliminary univariate analysis were included in the initial model, namely diabetes, obesity, significant lung involvement on admission, the CRP concentration, and all cell subsets with AUCs ≥ 0.65 (Table 4).

The four variables selected automatically for the final model were diabetes, significant lung involvement on admission, increased (above the 90% sensitivity threshold) CD8+ T-cell NKG2A MFI, and obesity. The adjusted ORs for the association of these clinical parameters with disease progression were 40.9 for diabetes, 13.3 for lung involvement and 13.8 for obesity (Table 4). In the presence of clinical comorbidities, increased (>2.054) CD8+ T-cell NKG2A MFI showed a strong capacity to predict disease progression (adjusted OR, 14.2; Figure 2 and Table 4). The estimated probability of progression to severe disease reached 86.2% in the presence of the four variables that were included in the model. The probabilities of COVID-19 progression according to the final logistic model are depicted in Table 5.

## 4. Discussion

Our study investigated the association of the key immune cell profile with disease progression in patients with hypertension who required hospitalization for COVID-19. The main findings can be summarized as follows: (i) reduced total lymphocyte count, CD4+ and CD8+ T cell count and CD4/CD8 ratio; and increased CD8+ HLA-DR MFI, and CD8+ NKG2A MFI on admission, were associated with progression to severe COVID-19; (ii) according to our logistic regression model, increased CD8+ NKG2A MFI at hospital admission, in combination with some clinical variables, including obesity and diabetes, is associated with a high risk of COVID-19 progression in these patients.

Clinical comorbidities, particularly diabetes, hypertension, and obesity, have been associated with COVID-19 severity in several studies, as we discussed in a previous work [11]. A chest CT examination has also been proposed for the prediction of clinical outcomes of COVID-19 [23,24]. In our previous study, we demonstrated that interleukin-10 and interleukin-12 (p70) levels, in combination with clinical variables, at hospital admission are key biomarkers associated with an increased risk of disease progression in hypertensive patients with COVID-19, and we proposed a biomarker-based approach to improve the prediction of this risk [11]. In this study, to improve the understanding of the associations of clinical variables and immune cell profiles with the risk of COVID-19 progression, we also used a forward stepwise predictive model, which combined immune cell subsets with clinical variables to predict the probability of progression to severe COVID-19.

Similar to our findings, several authors have reported reduced populations of lymphocytes, including T, B, and NK cells, and increased neutrophil counts in blood in patients with COVID-19, with more pronounced changes seen in severe cases [12,15,16,25,26]. Cell-mediated immunity plays a vital role in the immune response against viral infections, and interactions between the innate and adaptive immune systems are key parts of an effective host response. The functional impairment of cytotoxic lymphocytes and NK cells is associated with SARS-CoV-2 persistence [26].

Zheng et al. [17] proposed that COVID-19 progression is related to the functional exhaustion of T cells with the increased expression of NKG2A. NKG2A is an inhibitory receptor that can induce NK and T cell exhaustion, suppressing the cytotoxic activity of these immune cells and promoting viral spread during a variety of chronic viral infections [27,28]. Cytokine levels of IL-6 and IL-10, which are markedly increased in patients with COVID-19, can elicit the upregulation of NKG2A expression, potentializing its inhibitory role [25]. We previously observed increased levels of these cytokines in hypertensive patients with COVID-19 progression to severe disease [11]. Cho et al. showed that IL-6 and IL-10 enhanced CD94/NKG2A expression in naive CD8+ T cells [29], and different authors have demonstrated that higher levels of IL-10 are associated with increased NKG2A expression in viral infections [30,31,32]. In line with these findings, our results showed that in hypertensive patients, NKG2A also emerges as an important feature in patients with COVID-19 who progressed to severe disease.

The immune cell profile that was observed in this study corroborates the hypothesis that COVID-19 progression is related to T-cell exhaustion with increased NKG2A expression. These findings contribute to our understanding of how SARS-CoV-2 infection may break down the antiviral immune response at an early stage through this mechanism. These immune cell subsets might be useful for a better understanding and prediction of the progression to severe COVID-19. These findings are relevant, as they support the potential use of the anti-NKG2A monoclonal antibody monalizumab in patients with COVID-19. The use of this drug to control tumor growth by restoring CD8 + T and NK cell functions is currently under investigation [19]; phase-2 clinical trials have revealed no significant side effects [16,33].

This study has some limitations. Blood samples were collected at a mean of 2.7 days after hospitalization, and the median interval between symptom onset and hospital admission was 5.0 days. In addition, our population consisted only of hypertensive patients taking ACEIs/ARBs; the exclusion of patients with severe hypertension and high-risk clinical presentations (i.e., hemodynamic instability or mechanical ventilation requirement on admission) may limit the generalizability of our results. Still, the widespread use of ACEIs/ARBs in the hypertensive population and the multicentric nature of the study might help to ensure the good external validity of the findings. Patients’ previous use of glucocorticoids could have affected our analysis; however, only five (3.2%) of the patients were using this drug before hospitalization. Additional studies are needed to validate the results that were obtained in this work in more heterogeneous populations of hypertensive patients and to evaluate the applicability of the proposed model to non-hypertensive COVID-19 populations.

## 5. Conclusions

This study demonstrated that increased NKG2A expression among CD8 cells, in combination with some clinical variables (obesity, diabetes, and extensive lung involvement), could become a biomarker of COVID-19 progression. These findings reinforce the hypothesis that T-cell exhaustion, characterized by increased NKG2A expression, is a crucial mechanism of breaking the antiviral immune response in COVID-19. These data may have direct therapeutic implications for the future.

## Figures and Tables

**Figure 1 jcm-11-03713-f001:**
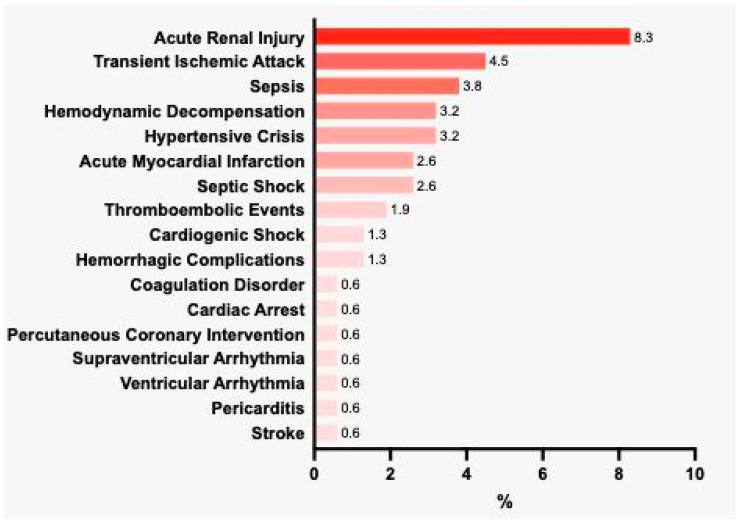
Main complications occurring during hospitalization (%).

**Figure 2 jcm-11-03713-f002:**
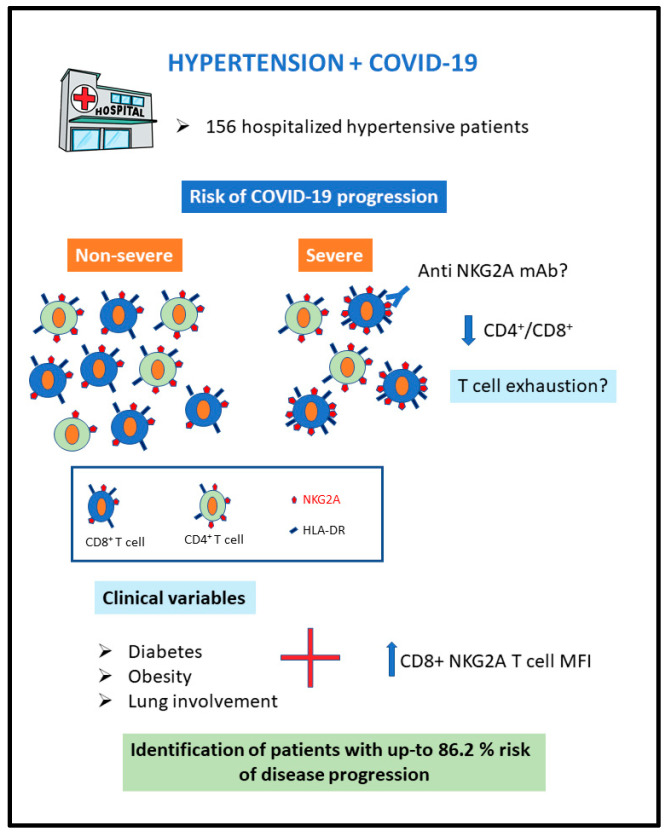
Model for the prediction of the risk of COVID-19 progression in the presence of increased CD8+ T-cell NKG2A MFI and clinical variables. HLA-DR, human leukocyte antigen DR isotope; mAb, monoclonal antibody; MFI, mean fluorescent intensity; NKG2A, natural killer group 2-member A.

**Table 1 jcm-11-03713-t001:** Baseline patient characteristics according to progression to severe disease *.

Clinical Conditions	Total	Non-Severe Cases (*n* = 145/156)	Severe Cases * (*n* = 11/156)	Fisher’s Exact Test *p*-Value
	*n*	*n* **(%)**	*n* **(%)**	
Sex				
Male	102	93 (91.2)	9 (8.8)	0.332
Female	54	52 (96.3)	2 (3.7)	
Age				
<60 years old	108	100 (92.6)	8 (7.4)	1.000
60 and older	48	45 (93.8)	3 (6.3)	
Signs of pulmonary involvement				
O2 sat > 93% and CT ≤ 50% †	124	121 (97.6)	3 (2.4)	<0.001
O2 sat ≤ 93% or CT > 50%	32	24 (75.0)	8 (25.0)	
Obesity				
No (BMI < 30 kg/m²)	75	74 (98.7)	1 (1.3)	0.010
Yes (BMI ≥ 30 kg/m²)	81	71 (87.7)	10 (12.3)	
Diabetes				
No	116	115 (99.1)	1 (0.9)	<0.001
Yes	40	30 (75.0)	10 (25.0)	
Asthma/COPD				
No	153	142 (92.8)	11 (7.2)	1.000
Yes	3	3 (100.0)	0 (0.0)	
Dyslipidemia				
No	131	122 (93.1)	9 (6.9)	0.690
Yes	25	23 (92.0)	2 (8.0)	
Coronary artery disease				
No	153	142 (92.8)	11 (7.2)	1.000
Yes	3	3 (100.0)	0 (0.0)	

BMI, body mass index; COPD, chronic obstructive pulmonary disease; CT, computed tomography; O2 sat, oxygen saturation on room air. * Modified World Health Organization (WHO) Ordinal Scale for Clinical Improvement scores of 6–8. † Extent of lung involvement on initial chest CT scan estimated by visual assessment performed by a radiologist.

**Table 2 jcm-11-03713-t002:** Biomarker levels according to disease progression (WHO score §).

	All (*n* = 156)			Non-Severe Cases † (*n* = 145)			Severe Cases ‡ (*n* = 11)		
	Median	Mean	SD	Median	Mean	SD	Median	Mean	SD
CRP, mg/L	2.37	4.62	5.38	2.12	4.14	5.01	11.49	10.97	6.25
Total lymphocyte count, ×10^−6^/L	1395	1521	758	1450	1529	670	800	1416	1559
Total monocyte count, ×10^−6^/L	230	369	383	230	357	323	100	525	863
Total B lymphocyte count, ×10^−6^/L	107	153	161	105	149	145	126	209	308
Total B lymphocytes %	8.0	10.2	8.5	7.8	9.8	8.1	15.8	15.3	11.1
NK cell count, ×10^−6^/L	159	199	180	159	197	175	137	217	246
NK cells, % *	10.8	14.3	11.2	10.6	14.1	11.4	16.6	15.9	8.1
NK-NKG2A, % **	30.7	34.1	16.5	30.6	33.8	16.4	39.2	37.7	18.0
NK-NKG2A MFI **	2.039	2.209	831	2.019	2.200	840	2.200	2.325	725
Total T lymphocyte count, ×10^−6^/L	1.031	1.150	633	1.088	1.165	591	520	954	1.059
Total T lymphocytes, %	78.0	74.3	14.8	78.2	74.9	14.2	64.9	65.7	19.9
CD4+ T cell count, ×10^−6^/L	549	651	443	564	660	406	241	522	805
CD4+ T cells, %	56.5	54.8	14.8	56.9	55.6	14.5	40.7	44.4	15.6
CD8+ T cell count, ×10^−6^/L	377	433	255	381	437	254	310	379	280
CD8+ T cells, %	37.7	39.0	14.2	36.9	38.4	13.9	42.6	46.8	16.5
CD4/CD8 *	1.51	1.76	1.06	1.55	1.79	1.05	0.73	1.25	1.05
CD8+ CD38+ T cells, % *	8.7	14.4	15.7	8.1	14.4	16.2	17.4	14.4	8.2
CD8+ CD38+ T cells MFI *	4035	4463	2453	4067	4511	2533	3893	3840	626
CD8+ HLADR+ T cells, % *	15.3	20.3	18.8	14.9	20.1	18.5	19.6	22.7	23.5
CD8+ HLADR+ T cells MFI *	939	1282	944	909	1252	936	1341	1678	1000
CD8+ NKG2A+ T cells, %	4.6	8.0	8.7	4.7	8.0	8.8	4.2	7.3	7.8
CD8+ NKG2A+ T cells MFI	2153	2154	804	2137	2130	819	2293	2480	460
CD8+ HLADR+ CD38- T cells, % *	7.9	14.2	16.2	8.2	14.0	15.6	7.2	16.6	23.0
CD8+ HLADR+ CD38+ T cells, % *	3.6	6.1	6.7	3.5	6.1	6.9	4.6	6.1	4.3
CD8+ HLADR- CD38+ T cells, % *	4.2	8.3	12.5	4.2	8.3	12.9	8.7	8.3	6.0
CD8+ HLADR- CD38- T cells, % *	76.6	71.4	21.3	76.6	71.6	21.3	75.4	69.0	21.7

CRP, C-reactive protein; HLA-DR, human leukocyte antigen DR isotope; MFI, mean fluorescent intensity; NK, natural killer; NKG2A, natural killer group 2-member A. § Modified World Health Organization Ordinal Scale for Clinical Improvement. * Missing for one patient; ** missing for two patients. † Non-severe, scores of 3–5; ‡ Severe, scores of 6–8.

**Table 3 jcm-11-03713-t003:** Associations of biomarkers with disease progression by areas under ROC curves.

	Area under Curve	*p*-Value	Cut-off for 90% Sensitivity
**CRP**	**0.827**	**<0.001**	**2.73**
**CD4+ T cells, % ***	**0.718**	**0.016**	**64.8**
**Total lymphocytes ***	**0.702**	**0.026**	**2100**
**CD4/CD8 ***	**0.692**	**0.034**	**3.23**
**CD8+ HLADR T cell MFI**	**0.670**	**0.060**	**624**
**CD8+ T cells, %**	**0.659**	**0.079**	**22.2**
**CD8+ NKG2A T cell MFI**	**0.650**	**0.097**	**2054**
Monocytes	0.644	0.112	**
B lymphocytes	0.643	0.115	**
T lymphocytes	0.636	0.132	**
NK cells	0.601	0.265	**
CD8+ HLADR- CD38+ T cells	0.597	0.283	**
CD8+ CD38+ T cells	0.592	0.312	**
CD8+ HLADR+ CD38+ T cells	0.585	0.350	**
NK-NKG2A cells	0.570	0.438	**
NK-NKG2A cell MFI	0.567	0.459	**
CD8+ NKG2A+ T cells	0.554	0.549	**
CD8+ CD38+ T cell MFI	0.551	0.577	**
CD8+ HLADR- CD38- T cells	0.548	0.596	**
CD8+ HLADR+ T cells	0.541	0.651	**
CD8+ HLADR+ CD38- T cells	0.534	0.707	**

CRP, C-reactive protein; HLA-DR, human leukocyte antigen DR isotope; MFI, mean fluorescent intensity; NK, natural killer; NKG2A, natural killer group 2-member A. * For total lymphocytes, CD4+ T cells and CD4/CD8, reduced values are predictors of disease progression. ** Not calculated due to lack of statistical association at 10% significance level. Bold values indicate AUC ≥ 0.650.

**Table 4 jcm-11-03713-t004:** Forward stepwise logistic regression model for the prediction of COVID-19 progression to severe disease.

Variables in the Equation		β	*p*-Value	Odds Ratio
Step 1	Diabetes	3.64	0.001	38.0
Step 2	Diabetes	3.37	0.002	29.2
	Lung involvement †	2.29	0.003	9.9
Step 3	Diabetes	3.74	0.001	42.0
	Lung involvement	2.39	0.006	10.9
	CD8+ NKG2A T cell MFI > 2054	2.78	0.020	16.0
Step 4 (Final model) *	Diabetes	3.71	0.002	40.9
	Lung involvement	2.59	0.008	13.3
	CD8+ NKG2A T cell MFI > 2054	2.65	0.037	14.2
	Obesity	2.63	0.040	13.8

* Constant = −9.746; Hosmer–Lemeshow test, *p* = 0.948. † Significant lung involvement (oxygen saturation ≤93% or >50% lung involvement on computed tomography examination) on admission. MFI, mean fluorescent intensity; NKG2A, natural killer group 2 member A.

**Table 5 jcm-11-03713-t005:** Probabilities of COVID-19 progression according to the final logistic model.

	CD8+ NKG2A+ T Cell MFI	
	≤2054 *	>2054
No clinical risk factor †	0.0%	0.1%
Diabetes only	0.2%	3.3%
Obesity only	0.1%	1.1%
Lung involvement only ‡	0.1%	1.1%
Diabetes + obesity	3.2%	31.9%
Diabetes + lung involvement	3.1%	31.1%
Obesity + lung involvement	1.1%	13.2%
Diabetes + obesity + lung involvement	30.6%	86.2%

* Cut-off for 90% sensitivity. † Clinical risk factors included in the model are diabetes, obesity, and significant lung involvement on admission. ‡ Oxygen saturation ≤93% or >50% lung involvement on computed tomography examination. MFI, mean fluorescent intensity; NKG2A, natural killer group 2 member A.

## Data Availability

The data presented in this study are available on request from the corresponding author.

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
