# Peer review of "NKG2A Expression among CD8 Cells Is Associated with COVID-19 Progression in Hypertensive Patients: Insights from the BRACE CORONA Randomized Trial"

_jcm, 2022, doi:10.3390/jcm11133713_

Round 1
Reviewer 1 Report
Dear Editor,
Moll-Bernades and colleagues conducted a translational research project exploring the key immune cell profile in patients with history of hypertension, who required hospitalization for COVID-19. 156 patients were included in this study. Among other findings, the researches reported that NKG2A expression among CD8 cells was associated with progression to severe COVID-19.
I would like to thank you for the opportunity to review this work. I would also like to commend the authors. Please see my comments below:
General comment:
· The main limitation of the study is the small population (N=156), although some one can argue that 156 patients is not a low number for a translational research project. The authors attempted to be as robust as possible in their statistical approach to mitigate this limitation.
Introduction:
- *The first sentence “Risk factors … COVID-19” needs some references. Among the plethora of available research, I would recommend PMID: 35160073, PMID: 33123973, PMID: 32422233, PMID: 33956286
- The background is nicely written and the objective is clearly defined.
Methods:
- 2.1: I would remove the word “diagnoses” after COVID-19 in line 4 of 2.1
- I would change “hypertensive patients” to “patients with hypertension”
- I would change “chronic … users” to “were on treatment with ACEI/ARB at baseline”
- I would change “coronary disease” to “coronary artery disease”
- Methods appear to be appropriate overall.
Results:
- I would remove the phrase “in the present study … patients”
- I would change “were obese” to “had obesity”
- Consider presenting data on in-hospital death (n, %)
Discussion:
- I would rephrase the 1st sentence of discussion.
- I would actually rephrase the whole 1st paragraph of discussion.
- Otherwise, discussion is appropriate.
Author Response
Responses to Reviewer
- General comment:
- The main limitation of the study is the small population (N=156), although some one can argue that 156 patients is not a low number for a translational research project. The authors attempted to be as robust as possible in their statistical approach to mitigate this limitation.
Response: We appreciate these comments from the reviewer.
- Introduction:
*The first sentence “Risk factors … COVID-19” needs some references. Among the plethora of available research, I would recommend PMID: 35160073, PMID: 33123973, PMID: 32422233, PMID: 33956286
Response: We thank the reviewer for these suggestions, and we have included the recommended references to better support our introduction.
The background is nicely written and the objective is clearly defined.
Response: We thank the reviewer for the acknowledgement.
Methods:
- 1: I would remove the word “diagnoses” after COVID-19 in line 4 of 2.1
Response: the word diagnoses was removed.
- I would change “hypertensive patients” to “patients with hypertension”
Response: We changed as recommended.
- I would change “chronic … users” to “were on treatment with ACEI/ARB at baseline”
- Response: We changed as recommended
- I would change “coronary disease” to “coronary artery disease”
Response: we corrected accordingly in methods section and also in Table1.
- Methods appear to be appropriate overall.
Response: Thank you for this comment
Response: We are very grateful for all this suggestion to improve the clarity of the paper and we did the modifications accordingly.
Results:
- I would remove the phrase “in the present study … patients”
Response: the phrase was removed as suggested.
- I would change “were obese” to “had obesity”
Response: We changed as suggested.
- Consider presenting data on in-hospital death (n, %)
Response: We thank the reviewer for these recommendations that improved our text. We included data regarding in-hospital death in the text (item 3.1 Outcomes)
Discussion:
- I would rephrase the 1st sentence of discussion.
- I would actually rephrase the whole 1st paragraph of discussion.
- Otherwise, discussion is appropriate.
Response: The first paragraph was entirely rephased, according to the reviewer’s recommendation and we are thankful for this suggestion.
Reviewer 2 Report
The paper is very interesting. The study was well conducted, and the manuscript was well-written and argued.
I believe this paper could be of interest to the readers as it highlights novel data that further explain molecular pathways triggered by SARS-CoV-2 infection and lead to severe forms of COVID-19 disease.
The introduction provides an adequate background and the discussion is complete and well-argued. Method and Results are clearly presented. The authors pointed out the limitations of the study which was correct.
Author Response
The authors thank the reviewer for the time they spent evaluating the manuscript and the comments.